# Structure of Ribosome-Inactivating Protein from *Mirabilis jalapa* and Its L12-Stalk-Dependent Inhibition of *Escherichia coli* Ribosome

**DOI:** 10.3390/toxins17120575

**Published:** 2025-11-28

**Authors:** Nanami Nishida, Yuki Ninomiya, Toru Yoshida, Takehito Tanzawa, Yasushi Maki, Hideji Yoshida, Hideaki Tsuge, Noriyuki Habuka

**Affiliations:** 1Faculty of Life Sciences, Kyoto Sangyo University, Kita-ku, Kyoto 603-8555, Japani2587235@cc.kyoto-su.ac.jp (Y.N.);; 2Department of Chemical and Biological Sciences, Faculty of Science, Japan Women’s University, 2-8-1 Mejirodai, Bunkyo-ku, Tokyo 112-8681, Japan; 3Institute for Protein Research, The University of Osaka, Suita, Osaka 565-0871, Japan; 4Department of Physics, Osaka Medical and Pharmaceutical University, Takatsuki 569-8686, Japanhideji.yoshida@ompu.ac.jp (H.Y.); 5Institute for Protein Dynamics, Kyoto Sangyo University, Kita-ku, Kyoto 603-8555, Japan; 6Center for Molecular Research in Infectious Diseases, Kyoto Sangyo University, Kita-ku, Kyoto 603-8555, Japan

**Keywords:** *Mirabilis* antiviral protein, ribosomal stalk, *E. coli* ribosomes, RNA *N*-glycosylase, sarcin-ricin loop

## Abstract

*Mirabilis* antiviral protein (MAP) is the type I ribosome-inactivating protein (RIP), which consists of an RNA *N*-glycosylase domain with no carbohydrate-binding domain. Unlike many RIPs, such as ricin or trichosanthin, which inactivate eukaryotic ribosomes, MAP also inactivates the *E. coli* ribosome by cleaving the *N*-glycosidic bond at A2660 of 23S ribosomal RNA. The structure of the wild-type MAP has not been revealed yet. Here, we expressed, purified, and crystallized the plural recombinant MAPs, including both E168Q and R171Q mutations (MAP-EQRQ) in *E. coli*, and determined the crystal structure of MAP-EQRQ at 2.1 Å resolution. According to the predicted structure with RNA (sarcin-ricin loop) and the mutant protein’s activities using quantitative RT-PCR, we showed that residue R171 at the active site of MAP is a key residue to form the stable complex with target adenine. Furthermore, we showed that MAP bound the C-terminal domains of eukaryotic P2-stalk as well as *E. coli* L12-stalk.

## 1. Introduction

Ribosome-inactivating proteins (RIPs) are a diverse family of toxic enzymes that catalyze the removal of the specific adenine residues from ribosomal RNA (rRNA *N*-glycosylase EC 3.2.2.22), thereby irreversibly inhibiting protein synthesis and leading to cell death [1,2]. The best-known RIPs are ricin from the castor bean (*Ricinus communis*), which depurinates explicitly the conserved adenine within the sarcin-ricin loop (SRL) of the 28S rRNA in eukaryotic ribosomes [3]. Because the SRL has a critical role in protein synthesis, which requires GTP hydrolysis together with elongation factors, its inactivation effectively halts protein synthesis [4].

Ricin belongs to the type II RIPs, which consist of an RNA *N*-glycosylase domain (A) and a carbohydrate-binding domain (B) in contrast, type I RIPs, which include trichosanthin (TCS) derived from *Trichosanthes kirilowii* [5], they have only an RNA *N*-glycosylase domain but lack a carbohydrate-binding domain. Interestingly, most RNA *N*-glycosylase domains of RIPs display strong specificity for eukaryotic ribosomes, having little to no effect on bacterial ribosomes.

*Mirabilis* antiviral protein (MAP) was identified in the roots of *Mirabilis jalapa* as a potent antiviral agent against the transmission of tobacco mosaic virus [6]. It is composed of 250 amino acids, having a combined molecular weight of 27,833, with 24% similarity with the ricin A chain (RTA) [7]. Both MAP and RTA inactivate the ribosomes by cleaving the *N*-glycosidic bond of a specific adenine of SRL, A4324 of rat 28S rRNA, in a hydrolytic fashion. Interestingly, MAP also shows a strong inhibitory effect against *E. coli* ribosomes and was shown to inactivate the *E. coli* ribosomes by cleaving the *N*-glycosidic bond at A2660 of 23S ribosomal RNA [8,9].

During translation on the ribosomes, a large amount of energy is necessary for protein synthesis, which is provided by GTP. The GTP hydrolysis occurs in the GTPase-associated center with the collaboration of the SRL and elongation factors, which are delivered by the eukaryotic ribosome P-stalk composed of a pentameric P-complex, with P0 and two copies of P1/P2 heterodimers [10]. It has been shown that the C-terminal domain (CTD) of ribosomal stalk proteins P1/P2 (P1-CTD, P2-CTD) is responsible for domain-specific binding of elongation factors to deliver them to the GTPase center [11,12]. On the other hand, *E. coli* ribosome stalks are made of L10 and L12 proteins, and the elongation factors bind to the CTD of L12 [13]. The crystal structure of the L12-CTD, made of 74 amino acids, was reported, but there is no similarity with eukaryotic P1/P2 CTD [14]. No interaction between RIP and the *E. coli* L10/L12 proteins has been reported.

RIP is assumed to hijack the CTD, be delivered to SRL RNA, and cleave the *N*-glycoside bond of the target adenosine [15,16,17,18,19]. For example, RTA-P protein interaction was reported to be essential for ribosome inactivating action, as yeast mutants with deletion of P1 and/or P2 have less depurination [15]. Although MAP and RTA were shown to depurinate naked rRNA, the RNA *N*-glycosylase activity is more than 100 times faster for rRNA within an intact ribosome [9,20], suggesting that the ribosomal proteins increase in susceptibility of rRNA toward RIPs.

Complex structures of RIP, such as TCS (2JDL) and RTA (5GU4), and the CTD consensus motif were reported [18,21], in which TCS interacts with a part of the C11-P2 (SDDDMGFGLFD), and RTA interacts with a part of the C6-P2 (GFGLFD). Both TCS and RTA bind the CTD consensus motif in a similar stalk binding region. However, their binding direction and their binding mode are different. The DDD sequence in the CTD forms favorable charge-charge interactions with positively charged TCS residues. The GFGLFD sequence of the CTD is inserted in a hydrophobic pocket of RTA [18,22]. The binding mode of the stalk CTD itself might be flexible among the RIPs.

Generally, RIPs have evolved to be near-perfect catalysts for eukaryotic ribosomes. Though MAP has a unique activity that shows inhibitory activity to *E. coli* ribosomes, it is an open question how MAP exhibits potent RNA *N*-glycosylase activity against *E. coli* ribosomes. Furthermore, the structure of the wild-type MAP has not been revealed yet. To address these questions, we expressed and purified three recombinant MAPs, including both E168Q and R171Q mutations (namely MAP-EQRQ) in *E. coli*. Then, we crystallized and determined the crystal structure of MAP-EQRQ. Using quantitative RT-PCR, we examined the inhibition activity of wild-type MAP and its mutants, E168Q, R171Q, and MAP-EQRQ.

In addition, we discussed that residue R171 in the active site of MAP is a key residue for forming the stable complex with RNA (SRL). Finally, using this MAP-EQRQ, we showed that MAP bound both eukaryotic and *E. coli* ribosomal stalk peptides, whereas recombinant TCS bound only the eukaryotic stalk peptide. This suggests that MAP inhibition is dependent on the *E. coli* ribosomal stalk.

## 2. Results

### 2.1. Determination of the Crystal Structure of the MAP-EQRQ Mutant

Because of the MAP effect on the *E. coli* ribosomes, wild-type MAP (wt-MAP) was difficult to produce by the *E. coli* system. It has been shown that RTA residues E177 and R180 play a crucial role in the hydrolysis of *N*-glycoside bonds by stabilizing the transition state during the catalysis of the depurination reaction [23,24]. As these two residues were conserved in MAP (Figure 1), the inactivated recombinant MAP-EQRQ was expressed, purified, and crystallized. X-ray diffraction data were collected at KEK BL5a, and the phase was determined using the molecular replacement (MR) method. A predicted structure by AlphaFold [25] was used for the initial search model of the MR. As a result, there were four molecules of MAPs in the asymmetric unit, which belong to the C2 space group. The final resolution is 2.1 Å, and R-free values were 19.4% and 23.9%, respectively (Table 1). Although many RIP structures have already been solved, we could not obtain the MR solution using their structures as search models. RMSD values against the final structure of MAP (C mol) are 0.962 Å (TCS:2jdl:149 pruned atom pairs), 1.051 Å (RTA:5gu4:150 pruned atom pairs), and 0.443 Å (Initial model by AlphaFold: 249 pruned atom pairs), respectively.

### 2.2. Overall Structure of MAP and Its Active Site

There are four molecules (A, B, C, and D) in the asymmetric unit, which are formed by a dimer of dimers (Figure 2). The two similar fold axes connected each dimer (AB and CD). The 2-fold symmetric dimer was formed by residues α7 (Y174, D177, and K178), α10 (E238 and K241), and α1 (L19) in the α7α8-loop (E182 and E185) (Figure 1 and Figure 2). RMSD values are 0.328 Å (A and B), 0.291 Å (A and C), and 0.328 Å (A and D); thus, each monomer structure of MAP-EQRQ is basically the same. The structure of MAP was shown from the direction of the active site (Figure 3). Using protein blast, we searched for similar RIPs in already analyzed structures by RMSD values across all pairs. Bouganin: 3CTK (2.34 Å), Pokeweed Antiviral Protein: 1APA (2.53 Å) [28], PDL4 from *P. dioica* leaves: 2QES (2.66 Å). Even when comparing structures in the top 10 (with scores of 112 or higher), the RMSD with MAP is more than 1 Å, indicating that MAP belongs to a different group from the known RIPs. Additionally, the A chain of abrin (1ABR) and the RTA (2AAI) appear in a late position, which are type II RIPs.

The overall structures of both RTA and TCS were similar to that of the MAP. Notably, the active site comprises the conserved amino acid residues in these RIPs, including Y72, Y118, E168, and R171 of MAP (Figure 3). These conserved residues are also shown in the amino acid alignment using ESpript [27] (Figure 1) and suggest being essential for the RNA *N*-glycosylase activity of these RIPs.

### 2.3. RNA N-Glycosylase Activities by R171 and E168 Single Mutants of MAP

According to the previous report [9], RNA *N*-glycosylase activities of the MAP mutants EQRQ, E168Q, and R171Q are investigated (Figure 4A). After *E. coli* ribosomes were incubated with the mutants, rRNAs were extracted and treated with aniline under acidic conditions. The rRNA incubated with wt-MAP and E168Q mutant showed a specific band corresponding to 23S rRNA cleaved at A2660, whereas those with R171Q, or the EQRQ mutant, did not. To address the importance of the active-site residues R171 and E168 for MAP’s RNA *N*-glycosylase activity, we established an RT-PCR-based assay and quantified the amount of intact SRL sequence in the *E. coli* ribosomes (Figure 4B). According to the assay, approximately 98% of *E. coli* ribosomes (0.2 µM) were depurinated by wt-MAP at 0.050 µM, and approximately 90% by the single mutant E168Q at 5 μM (Figure 4B), whereas the double mutant MAP-EQRQ, as well as the single R171Q mutant, showed no inhibitory activity even at 5 µM (Figure 4B). The results indicate an important role for the R171 residue in the *N*-glycosylase action.

### 2.4. Arg171 of MAP Affects the Stability of the RNA (SRL) Binding

Though many RIPs and their complex structures with substrate analogs have been registered in the PDB, those of RIPs in complex with SRL RNA have not yet been reported. So far, the structures with substrate analogs such as formycin (RTA: PDB 3RTI), adenosine (TCS: PDB 1MRJ), or NADPH (TCS: PDB 1TCS) are available [29,30,31]. We used AlphaFold to predict the complex structures of the MAP and SRL RNA of *E. coli* ribosomes in order to clarify the active site features. That is how the active site of wt-MAP, as well as mutated MAP, interacts with SRL RNAs. The SRL sequence is conserved in *E. coli*, yeast, and rat as AGUACGAGAGGA, including GAGA-tetra-loop [32], in which the first adenine of the GAGA-loop (A2660 of *E. coli* ribosomes) is depurinated by MAP [9]. The conserved 12nt and wt-MAP or its mutants complex were predicted by AlphaFold [25]. The results yielded five predicted structures, which showed that wt-MAP bound uniformly to SRL RNA (Figure 5A). Furthermore, it should be noted that the target adenine flipped out and bound with R171 in all five structures, suggesting that R171 was an essential residue to make a stable complex with SRL (Figure 5B). The adenine position was the same as seen in the TCS-adenine complex (1QD2). On the other hand, in MAP E168Q, the adenine shows the same position as wt-MAP in two structures out of five. Interestingly, in MAP R171Q, adenine, A2662, was located at the same position in all five structures, but the direction of the adenine was totally different from that of wt-MAP. Furthermore, in MAP-EQRQ, there were conformational variations in adenine positions among the five predicted conformations of SRL (Figure 5A).

### 2.5. E. coli Stalk Binding Property of MAP

To address whether MAP binds both to the eukaryotic and *E. coli* stalk, we observed the binding properties between MAP-EQRQ and recombinant glutathione S-transferase (GST)-stalk peptides made of GST fusing the C-terminal peptide based on the eukaryotic ribosomal P2-protein SDDDMGFGLFD (11 aa) and KEESEESDDDMGFGLFD (17 aa) [22] and the *E. coli* L12-CTD (74 aa) [14], using TCS as a reference protein, which does not inactivate *E.coli* ribosomes (unpublished data). After incubation with both MAP and the GST-stalk peptides, complexes were trapped by Ni-resin via the hexa-histidine tag of MAP, and eluted proteins were analyzed, showing that MAP made complexes with the GST-stalk peptides of both eukaryotic and *E. coli* stalk peptides (Figure 6). On the other hand, recombinant TCS has a hexa-histidine tag bound only to that of the eukaryotic stalk peptide, not to that of the *E. coli* stalk peptide.

Interestingly, the theoretical molecular weight of MAP/TCS is larger than that of GST., i.e., MAP-EQRQ:GST. i.e., MAP-EQRQ: 257aa MW = 28,760, TCS: 253aa MW = 27,997, GST: 239aa; MW = 27,898, GST-11aa: 237aa; MW = 27,511, GST-17aa: 243aa; MW = 28,243, GST-74aa: 300aa; MW = 33,855. Even though MAP/TCS runs faster than GST in SDS-polyacrylamide gel electrophoresis, it shows a band separately from GST and its fusion protein.

### 2.6. Structural Property in the Stalk Binding Region of MAP

Peptide binding regions are common in RTA and TCS; however, the direction and mode of binding differ between them [18,21]. We impose the MAP structure with TCS/RTA with the eukaryotic stalk to see if there is any difference in these stalk-binding regions; in the stalk-binding region, crystal structures of MAP showed that the peptide binding region is hydrophobic and formed by Y174, Y192, W232, F234, and V243 (Figure 7). Among MAP and TCS/RTA, the structures of these sites differ broadly, such that MAP lacks the stalk-binding hydrophobic deep pocket, as seen in RTA. MAP has a shallow pocket as well as TCS; both MAP and TCS have hydrophobic properties in the lower half (Figure 7C,F). However, the surface property of the top half region is different between MAP and TCS (Figure 7B,E). In MAP, the feature of the upper half region is acidic, formed by E182 and E185 (Figure 7A,B). On the contrary, in TCS, the feature of the upper half region is basically formed by R174 and K177 (Figure 7D,E). In TCS, the DDD sequence in the CTD of the stalk peptide forms favorable charge-charge interactions with the TCS upper half basic region. These property differences in MAP suggest that MAP binds human stalk using a different binding mode than that of TCS. Furthermore, the specific feature of MAP might enable it to bind to the *E. coli* stalk.

## 3. Discussion

In this report, we pursued the following five issues. (1) We successfully expressed the inactive MAP mutant, MAP-EQRQ, in *E. coli*. (2) We solved the crystal structure of MAP-EQRQ at 2.1 Å resolution. In the past, MAP purified from the *Mirabilis jalapa* root was crystallized and reported [33], but its structure has not been solved by molecular replacement or any other phasing methods. As shown in this report, we expressed and purified an inactive homogeneous MAP-EQRQ, which gave good crystals. In addition, we used the predicted structure using AlphaFold as the initial model for the molecular replacement method to lead to the determination of the MAP-EQRQ structure; in both type 1 and type II RIPs, overall structures are similar, but there are some differences, especially in the stalk-binding region. RMSD between the initial AlphaFold model and the C molecule was 0.467 Å (across all 250 atom pairs). We confirmed that the low RMSD value is the key to solving the structure by MR. Especially, in this case, the search for four molecules in an asymmetric unit may also make it difficult to obtain the real solution. The obtained structure was similar to those of TCS and RTA and confirmed the mode of action by the conserved amino acid residues, such as Y72, Y118, E168, and R171 of MAP (Figure 1). (3) The RNA *N*-glycosylase activities of the mutant MAPs supported the importance of the active site R171 and E168 residues. As mutation R171Q shows no inhibition, R171 is an essential residue together with E168 to express the full inhibition activity of the *E. coli* ribosomes. (4) We predicted the complex structure of MAP and 12 nt SRL by AlphaFold. Up to now, there are no reports of RIPs in complex with SRL RNA. It suggested R171 is the essential residue that interacts with the adenine base and forms the MAP-RNA complex before the catalysis. (5) Finally, we showed that MAP binds to both the eukaryotic P2-stalk and the *E. coli* stalk (L12) CTDs, whereas TCS binds with only the eukaryotic P2-stalk CTD. It suggests the stalk-dependent inhibition of *E. coli* ribosomes by MAP.

RTA was reported to bind the eukaryotic P-stalk peptide using similar GST-fusion proteins [22]. Plural RIPs, including Pokeweed antiviral protein (PAP), were reported to inactivate *E. coli* ribosomes by cleaving the *N*-glycosidic bond of A2660 in their 23S rRNAs [34]. This is the first report indicating binding of RIP with the *E. coli* stalk L12 protein, which is a similar functional protein P1/P2 of the eukaryotic ribosomes.

When the ribosomes were deproteinated, RTA was reported to cleave the *N*-glycosidic bond of A2660 in the 23S rRNA [20]. Similarly, MAP also cleaved the bond of deproteinaized *E. coli* ribosomes but needed a much higher concentration [9]. In addition, TCS was reported to cleave the *N*-glycosidic bond of A2660 in 23S ribosome RNA when the stalk L12 protein of *E. coli* ribosomes was replaced by the eukaryotic P-stalk protein [35]. According to the observations, we assumed that the A2660 in *E. coli* 23S ribosome RNA is susceptible to RIPs, and the interaction of stalk L12 protein facilitates some RIPs activities to attack the *N*-glycosidic bond.

The antiviral properties of RIPs have been investigated for more than four decades. However, their antiviral mechanisms are still an open question. We expect that future studies on RIPs will shed light on this unknown mechanism.

## 4. Experimental Procedure

### 4.1. Purification, Cloning, and Expression

Unless otherwise specified, all chemicals and reagents used in the experiments were purchased by Fuji Film, Japan. All DNAs were synthesized by Thermo Fisher Scientific, Japan.

The MAP was extracted from the roots of *Mirabilis jalapa*. The squeezed solution from the root was precipitated using 90% saturated ammonium sulfate, and then it was dialysed against a 10 mM sodium phosphate buffer (pH 6.0) containing 5 mM 2-mercaptoethanol. Further MAP purification was achieved using cation-exchange chromatography (SP-Sepharose, Cytiva, Japan), followed by gel filtration chromatography (Superdex 75, Cytiva, Japan).

The synthesized MAP genes, including mutation E168Q and R171Q, as well as the TCS gene, were subcloned into pET21a within C-terminal hexa-histidine residues at their C-terminal ends and expressed in *E. coli* BL21(DE3) strain (Novagen, Japan). In the cases of single E168Q or R171Q, the OmpA-signal sequence was inserted between the initial Met codon and the first Ala codon of the MAP gene to secrete and accumulate the products in the culture media. The obtained recombinant proteins were purified with Ni-IMAC (Fuji Film, Japan), followed by gel-filtration using Superdex 75.

The recombinant glutathione-S transferase (GST) protein was obtained using the pGEX-4T1 plasmid harbored in the *E. coli* BL21 strain (Novagen, Japan). PCR was conducted using the PGEX-4T1 plasmid and DNA primer sets, GEX17F: GCGATGATGATATGGGCTTCGGCCTGTTTGATTAACTCGAGCGGCCGCATCGTGAC/GEX17R: GGCCGAAGCCCATATCATCATCGCTTTCTTCGCTTTCTTCTTTGGATCCACGCGGAACCAGATCCG and GEX11F: GATATGGGCTTTGGCCTGTTTGATTAACTCGAGCGGCCGCATCGTGACTGAC/GEX11R: CAGGCCAAAGCCCATATCATCATCGCTGGATCCACGCGGAACCAGATCCGG, followed by connection using the Gibson reaction, resulting in GST fused eukaryotic P-stalk peptides coding 11 amino acids (SDDDMGFGLFD) and 17 amino acids (KEESEESDDDMGFGLFD). Gene of the *E. coli* stalk L12-CTD peptide was cloned from *E. coli* W3110 strain using DNA primer sets: GEX72CF: GAAGTTGAAGTTAAATAACTCGAGCGGCCGCATCGTGACTGACTG/GEX72CR: CAGTCAGTCACGATGCGGCCGCTCGAGTTATTTAACTTCAACTTC, resulting in a 74 amino acid sequence (AAEEKTEFDVILKAAGANKVAVIKAVRGATGLGLKEAKDLVESAPAALKEGVSKDDAEALKKALEEAGAEVEVK). PCR was conducted using the PGEX-4T1 plasmid and DNA primer sets, GEX72NF: GGATCTGGTTCCGCGTGGATCCGCTGCTGAAGAAAAAACTG/GEX72NR: CAGTTTTTTCTTCAGCAGCGGATCCACGCGGAACCAGATCC. Then, the pGEX vector and *E. coli* stalk peptide gene were connected using the Gibson reaction. Similarly to GST protein, the GST-fused stalk peptides were expressed in *E. coli* BL21 strain. All the recombinant proteins were purified with glutathione-affinity chromatography, followed by gel filtration (size exclusion) chromatography using Superdex 75.

### 4.2. RIP and GST-Stalk Peptide Binding Assay

Fifty µg of the MAP-EQRQ or TCS were mixed with 50 µg of each GST-stalk in 1 mL of the buffer containing 10 mM Tris-HCl (pH 7.5), 150 mM NaCl, and waited for 30 min on ice. Twenty µL of the pre-equilibrated Ni-NTA-agarose was added to the mixture, then incubated for 30 min on ice. The resin was washed with 500 µL of 10 mM Tris-HCl (pH 7.5)-150 mM NaCl, followed by 500 µL of 20 mM imidazole in 10 mM Tris-HCl (pH 7.5)-150 mM NaCl. Proteins were eluted with 50 µL of 500 mM imidazole in 10 mM Tris-HCl (pH 7.5)-150 mM NaCl, then 10 µL of each sample was heat-denatured at 98 °C for 5 min in the presence of 2-mercaptoethanol and analyzed by SDS-polyacrylamide gel electrophoresis (10–20%).

### 4.3. Quantitative RT-PCR-Based RNA N-Glycosylase Assay

*E. coli* W3110 70S ribosomes at the final concentration of 200 nM were incubated with MAP purified from *Mirabilis* root or recombinant MAP in 50 μL of 25 mM Tris-HCl, pH 7.5, 25 mM KCl, 5 mM Magnesium Acetate at 37 °C for 30 min. RNA was extracted using NucleoSpin^TM^ (Takara-Bio, Japan). The RNA obtained was analyzed by RT-PCR using iTaq universal SYBR Green Rex^TM^ (Bio-Rad, Japan) and DNA primers: CGCTGGAGAACTGAGGGG and CAAGTTTCGTGCTTAGATGCTTT.

### 4.4. Crystallization, Data Collection, and Processing

Crystallization was performed using the hanging drop vapor diffusion method at 293 K with Crystal Screen and Crystal Screen Cryo (Hampton Research, Aliso Viejo, CA USA). Plate-like crystals were obtained under the condition crystal screen #30 (0.2 M Ammonium sulfate and 30% w/v Polyethylene glycol 8000). The crystals were cryoprotected by the mother liquor with glycerol. Diffraction data sets were collected using Dectris Pilatus3 S6M (Switzerland) on beamline KEK BL5A at 1.00 Å. Data processing was performed with XDS [36] (Table 1).

### 4.5. Structure Determination and Refinement

Initially, the structure of MAP (residues 1-250) was predicted using AlphaFold [25]. Using the AlphaFold predicted structure, a molecular replacement (MR) calculation was applied, and the solution revealed four molecules of MAP. Then, the model corrections were repeated using Coot [37] and PHENIX [38].

The final model includes four molecules of MAP (A:3-250, B1-256 including 6X His-tag at C-terminal, C:1-250, D:1-250), 13 SO_4_ molecules, and water molecules (Table 1). There are no outliers in the Ramachandran plot computed using PROCHECK [39] and MolProbity [40]. The coordinate was deposited in the Protein Data Bank with accession code 9X2O. All structural figures were drawn by PyMol (DeLano Scientific, Palo Alto, CA, USA) and ChimeraX [41].

### 4.6. Prediction of the MAP-RNA Binding Model Using AlphaFold

As described before, we calculated the apo MAP structure to use as an MR search model. We also predicted the structure of MAP with SRL of ribosomal RNA (12 nt: AGUACGAGAGGA). The cited results were checked to see any differences using not only wt-MAP, but also MAP-EQRQ, MAP-E168Q, and MAP-R171Q.

## Figures and Tables

**Figure 1 toxins-17-00575-f001:**
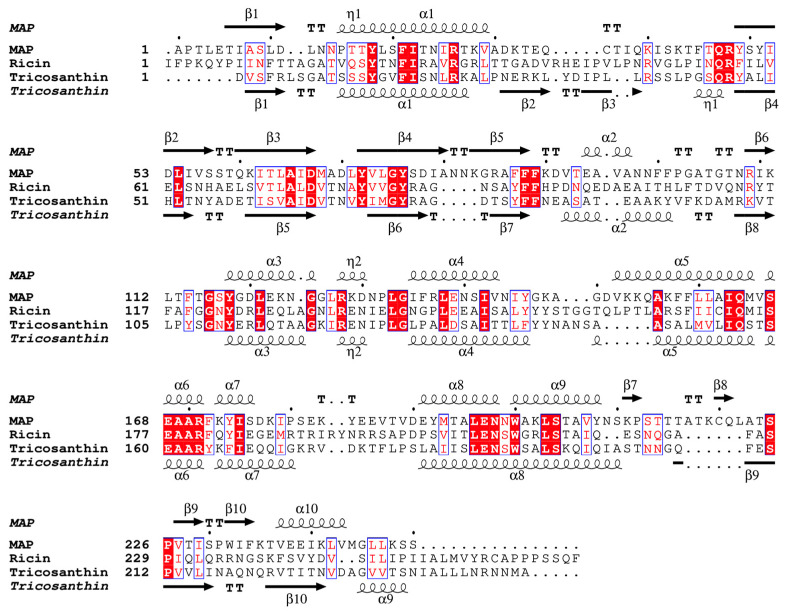
Sequence alignment of MAP, RTA (ricin A-chain), and TCS (trichosanthin) by Clustal Omega [26] and ESPript 3.03 [27]. The secondary structure was shown based on the crystal structure of MAP-EQRQ and TCS (1TCS). The η symbol indicates a 310-helix. β-strands are shown as arrows, strict β-turns as letters TT. There are four invariant amino acids in the active site: Y80, Y123, E177, and R180 in ricin. Y72, Y118, E168, and R171 in MAP.

**Figure 2 toxins-17-00575-f002:**
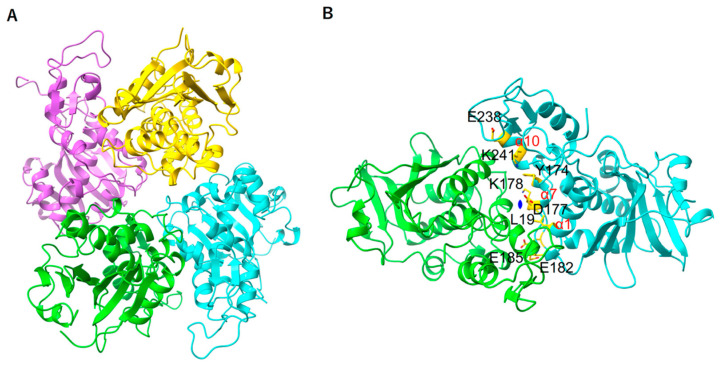
(**A**) MAP-EQRQ structure: four molecules (A (purple), B (yellow), C (green), and D (cyan)) in an asymmetric unit were shown. (**B**) MAP dimer structure (C and D) was shown. AB and CD dimer formation are the same. Pseudo two-fold symmetry was shown as blue. All captions belong to molecule D. Captions in molecule C are not included to clarify.

**Figure 3 toxins-17-00575-f003:**
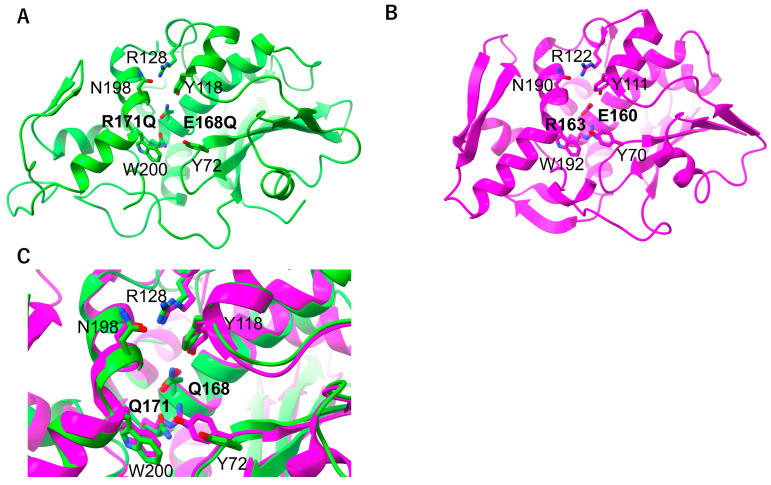
RNA *N*-glycosylase active site of RIPs (**A**) MAP-EQRQ structure (Green). Double mutations, E168Q and R171Q, were shown in bold. Other conserved residues were shown as fine print. (**B**) TCS (Purple) E160 and R163 were shown as bold, and other conserved residues were shown as fine print. (**C**) Close-up of the imposed structure of MAP-EQRQ and TCS.

**Figure 4 toxins-17-00575-f004:**
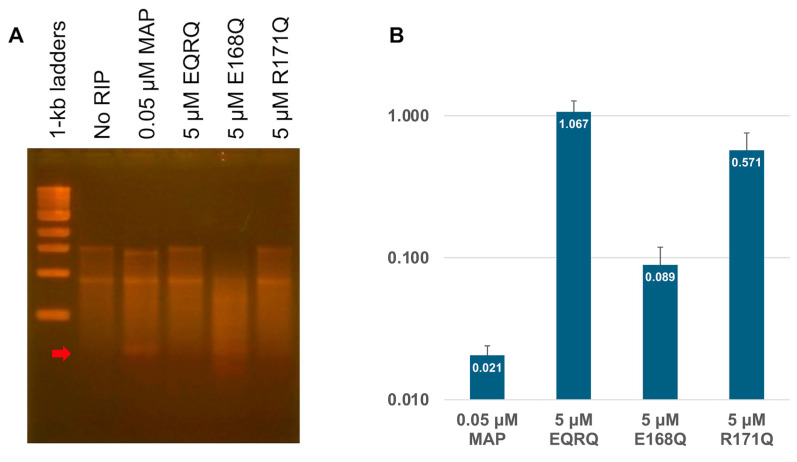
RNA-*N*-glycosylase assay of MAP and its mutants. (**A**) *E. coli* W3110 70S ribosomes (0.2 μM) were incubated with 0.05 μM MAP or each 5 μM of MAP-derivatives; E168Q/R171Q, E168Q, and R171Q in 50 μL of reaction buffer; 25 mM Tris-HC1 pH 7.5, 25 mM KC1, 5 mM MgOAc2 at 37 °C for 30 min. RNA was extracted utilizing NucleoSpin^®^. Aniline treatment was performed based on a previous report [9]. Obtained RNAs were incubated at 60 °C for 10 min, then applied to 2% agarose gel electrophoresis. The red arrow indicates the cleaved short 23S rRNA fragment at A2660 by the aniline treatment. (**B**) RT-PCR assay to quantify intact *E. coli* ribosome RNA in the SRL region. RT-PCR was performed using iTaq Universal SYBR green one-step system with primers; CGCTGGAGAACTGAGGGG and CAAGTTTCGTGCTTAGATGCTTT. The amount of intact ribosomal RNA was defined as 1.00 and then standardized using the quantity of 10- and 100-fold diluted samples as 0.1 and 0.01, respectively.

**Figure 5 toxins-17-00575-f005:**
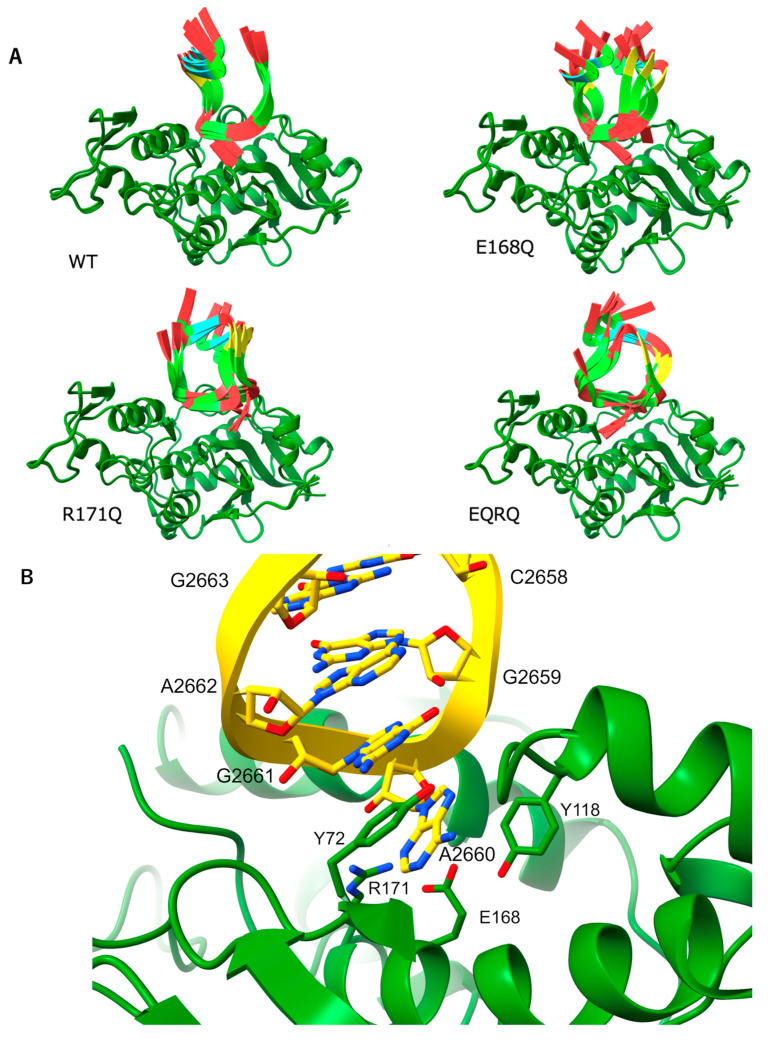
Predicted complex structure of MAP with SRL RNA by AlphaFold (**A**) Complex structures of wt-MAP or its mutants with SRL RNA (12 nt AGUACGAGAGGA: conserved region among *E.coli* and eukaryotic ribosome SRL RNA) (**B**) Closeup view of complex structures of wt-MAP, which is the representative structure in all five predicted structures. Adenine 2660 is stacked via π–π interactions by Y72 and trapped by R171. The distances between NH1 (NH2) and adenine N3 are 3.5 (3.2) Å.

**Figure 6 toxins-17-00575-f006:**
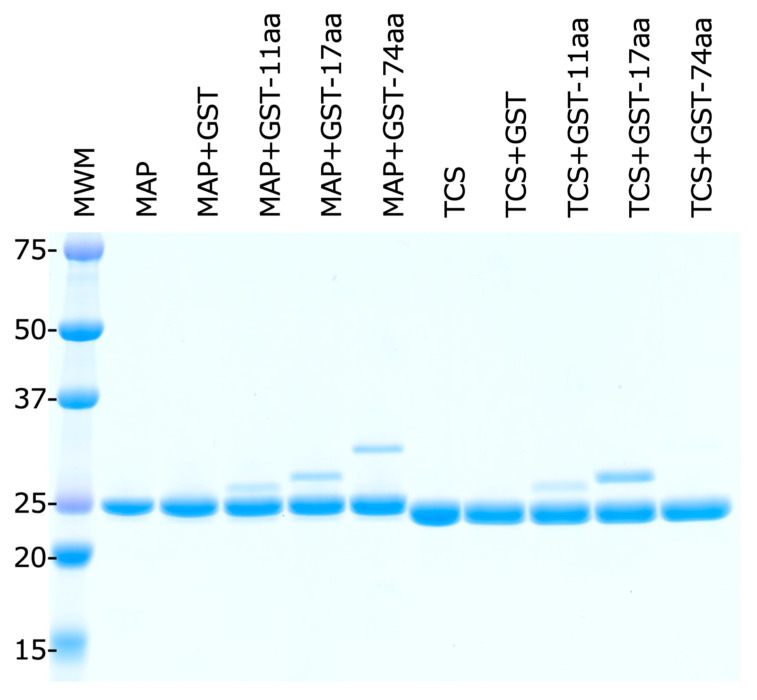
Stalk peptide interaction with RIPs. MAP: MAP-EQRQ, TCS (trichosanthin), GST-11aa, -17aa, and -74aa: glutathione S-transferase (GST) fused human P2-stalk C-terminal peptide sequences (11aa: SDDDMGFGLFD and 17aa: KEESEESDDDMGFGLFD) and *E.coli* stalk C-terminal domain made of 74aa, respectively. Fifty μg of the MAP-EQRQ or TCS were mixed with 50 μg of each GST-stalk, and then the bound proteins were trapped by and eluted from Ni-resin.

**Figure 7 toxins-17-00575-f007:**
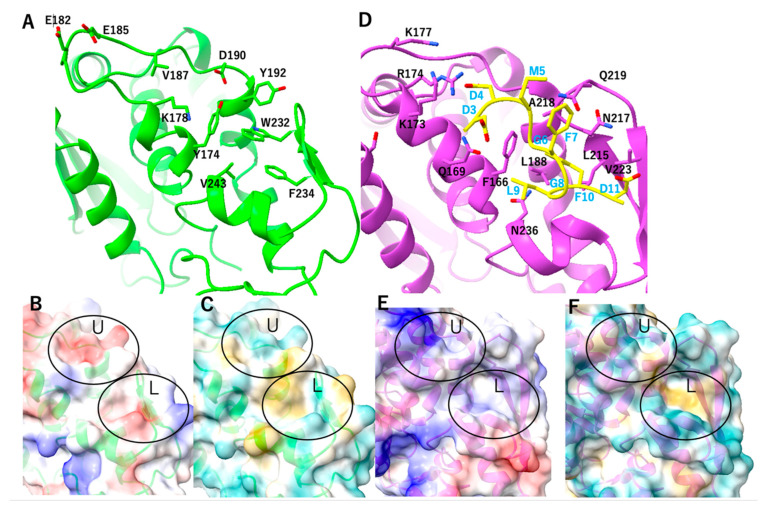
Stalk binding site of RIPs (**A**) Close up view of MAP-EQRQ (green) (**B**) Electrostatic surface of MAP-EQRQ (red: acidic blue: basic) (**C**) Hydrophobic surface of MAP-EQRQ (yellow: hydrophobic cyan: hydrophilic) (**D**) Closeup view of TCS (purple) with human 11aa stalk peptides (yellow) (**E**) Electrostatic surface of TCS without the stalk (red: acidic blue: basic) (**F**) Hydrophobic surface of TCS without the stalk (yellow: hydrophobic cyan: hydrophilic): U and L show the upper and the lower region of the stalk binding site as described in main text, respectively.

**Table 1 toxins-17-00575-t001:** X-ray data collection and refinement statistics.

	MAP-EQRQ
PDB ID	9X2O
**Data collection**	
Space group	*C* 1 2 1
Cell dimensions	
a, b, c (Å)	214.3, 60.7, 79.3
β (°)	109.0
Wavelength (Å)	1.0000
Resolution	43.9–2.1 (2.22–2.15)
*R_meas_*	0.300 (1.141)
*R_pim_*	0.115 (0.436)
CC_1/2_	0.984 (0.529)
*I/σI*	7.4 (2.0)
Completeness (%)	99.87 (99.57)
Redundancy	3.4 (3.4)
**Refinement**	
No. reflections	56,406
*R_work_/R_free_*	0.019/0.024
No. waters	8614
*B* factors (Å^2^)	
Protein	18.51
Ligands	37.71
Water	26.47
r.m.s. deviations	
Bond lengths (Å)	0.007
Bond angles (Å)	0.84
Ramachandran plot	
Favored (%)	97.38
Allowed (%)	2.62
Outliers (%)	0.00

## Data Availability

The coordinates and structure factors for the MAP-EQRQ structure was deposited in the PDB with the accession number 9X2O (https://www.rcsb.org/structure/9X2O, accessed on 17 October 2025). The structure pdb_00009x2o (PDB ID 9X2O) (Deposition ID D_1300063774) and the associated experimental data will be released on 3 December 2025.

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
