# Peer review of "Structure of Ribosome-Inactivating Protein from Mirabilis jalapa and Its L12-Stalk-Dependent Inhibition of Escherichia coli Ribosome"

_toxins, 2025, doi:10.3390/toxins17120575_

Round 1
Reviewer 1 Report
Comments and Suggestions for Authors
The authors solved the crystal structure of the inactive form of mirabilis antiviral protein (MAP), which is unique from other RIPs, because it targets on both eucaryotic and procaryotic ribosome. According to the structural details of MAP complemented with biological assays provided, key residue was found in 23S SRL binding. This work is interesting and worth to publish. The authors are requested to address the following questions before this manuscript can be accepted.
- With the structure solved, the authors are recommended to discuss the regions that are unique to MAP but not other RIPs and give reasons on why MR using other RIP structures was not successful.
- For the figure 4A, it is recommended to repeat the experiment to make an outstanding b-fragment and intact RNA band of 23S. It seems that degradation existed in E168Q group which decreases the amount of accumulated b-fragment. Second, since the wild-type MAP was purified from Mirabilis root, the purification methods should be clearly stated on how to avoid contaminants.
- Label the y-axis in figure 4B.
- A2660 in figure 5 should be inserted between two tyrosine rings. The authors are recommended to indicate this interaction.
- Does MAP have similar affinity to the L12 and P2-stalk CTD?
- The authors are recommended to discuss why MAP can depurinate both eukaryotic and prokaryotic ribosome.
Author Response
Reviewer 1
The authors solved the crystal structure of the inactive form of mirabilis antiviral protein (MAP), which is unique from other RIPs, because it targets on both eucaryotic and procaryotic ribosome. According to the structural details of MAP complemented with biological assays provided, key residue was found in 23S SRL binding. This work is interesting and worth to publish. The authors are requested to address the following questions before this manuscript can be accepted.
We appreciated the reviewer 1 for all important suggestions.
- With the structure solved, the authors are recommended to discuss the regions that are unique to MAP but not other RIPs and give reasons on why MR using other RIP structures was not successful.
Thank you for the comments.
We revised as follows in L106. “The structure of MAP was shown from the direction of the active site (Figure 3). Using protein blast, we searched for similar RIPs in already analyzed structures by RMSD values across all pair. Bouganin: 3CTK (2.34Å), Pokeweed Antiviral Protein: 1APA (2.53Å)[26], PDL4 from P. dioica leaves: 2QES (2.66Å). Even when comparing structures in the top 10 (with scores of 112 or higher), the RMSD with MAP is more than 1 Å, indicating that MAP belongs to a different group from the known RIPs. Additionally, the A chain of abrin (1ABR) and the RTA (2AAI) appear in a late position, which are type II RIP.
We added next sentences in the discussion. “In both type 1 and type II RIPs, overall structures are similar, but there are some differences in especially the stalk-binding region. RMSD between the initial AlphaFold model and C molecule was 0.467 angstroms (across all 250 atom pairs). We confirmed that the low RMSD value is the key to solve the structure by MR. Especially, in this case, four molecules search in as asymmetric unit may be also hard to obtain the real solution.”
- For the figure 4A, it is recommended to repeat the experiment to make an outstanding b-fragment and intact RNA band of 23S. It seems that degradation existed in E168Q group which decreases the amount of accumulated b-fragment. Second, since the wild-type MAP was purified from Mirabilis root, the purification methods should be clearly stated on how to avoid contaminants.
Thank you for your suggestion, similar suggestion is obtained from another reviewer, and picture is going to be replaced better one. However, the aniline treatment seems to be drastic for RNA molecules and often cause their degradation. We have done more than ten times to get clear picture without success. So far, the cleaved short 23S rRNA fragment was visible, then we pursued RT-PCR which compensated the aniline-experiment and showed quantities of depurinated 23S rRNA without aniline treatment.
Purification scheme of the wt-MAP was added to Experiment Procedure as follows: The wt-MAP extract was squeezed out from Mirabilis jalapa root, and precipitation obtained with 90% ammonium sulfate saturation was resuspended and dialyzed against 10 mM sodium phosphate buffer (pH 6.0)-5 mM 2-mercaptoethanol. MAP was further purified using cation-exchange (SP-Sepharose) chromatography followed by gel filtration using Superdex 75.
- Label the y-axis in figure 4B.
1.000, 0.100 and 0.010 are indication of Y-axis. We added an explanation of each number in the legend.
- A2660 in figure 5 should be inserted between two tyrosine rings. The authors are recommended to indicate this interaction.
We changed to the new figure 5B.
- Does MAP have similar affinity to the L12 and P2-stalk CTD?
Based on the intensity of bound GST-fusions, we did not see any difference between the L12 and P2-stalk CTD fusions.
- The authors are recommended to discuss why MAP can depurinate both eukaryotic and prokaryotic ribosome.
We have discussed shortly in the last paragraph of Discussion section. Not only MAP but also RTAs are able to depurinate the A2660 of E. coli 23S rRNA when ribosomal protein was eliminated. The key to cleave the site is assumed whether RIP is delivered by the stalk or not. We are still trying to have more clues to assure the hypothesis.

Reviewer 2 Report
Comments and Suggestions for Authors
The authors study the structure of MAP (Mirabilis Antiviral Protein) mutants and how they interact with the sarcin-ricin loop and ribosomal stalk proteins of eukaryotic and prokaryotic ribosomes, comparing it with trichosantin and ricin. MAP, tricosanthin and ricin are ribosome-inactivating proteins (RIPs) that have N-glycosylase activity on the sarcin-ricin loop (SRL) of the major ribosomal RNA. The SRL forms part of the GTPase-associated centre (GAC), which is the landing platform for translational GTPases. RIPs are used in cancer and antiviral therapy, and in agriculture for the construction of transgenic plants resistant to viruses, fungi and insects.
The topic is interesting, and the experiments provide new information that attempts to explain why some RIPs inhibit protein synthesis in bacteria; however, it is not well written, which makes it difficult to understand. Therefore, before being accepted, it needs to be rewritten.
Abstract:
Line 8: change “RNA N-glycosidase” by “RNA N-glycosylase”.
Lines 15 and 73: the expression “active site R171” is confusing.
Introduction:
In order that the reader may understand the purpose of this work, a systematic presentation of the state of the art is required, including (1) a definition of RIPs and types of RIPs, (2) enzymatic activity, (3) the function of the sarcin-ricin loop, (4) the function and composition of the GTPase-associated centre (GAC), and (5) the difference between the GAC of eukaryotes and prokaryotes.
Lines 21–22, 28, 30, …: change “rRNA N-glycosidase” by “rRNA N-glycosylase”
Lines 54-55: I am not sure whether it is clear that C11-P2 and C6-P2 are part of the CTD, but it would be useful to indicate this.
Line 61: Only L10 and L12, or also L11?
Results:
Section 2.1: This section could be improved. There are four invariant amino acids in the active site of ricin: Y80, Y123, E177, and R180. The tyrosines form a sandwich with adenine, and E177 and R180 are the amino acids directly involved in catalysis. They are incorrectly numbered in Figure 1. In addition, the legend for Figure 1 should explain the meaning of the symbols, letters and drawings.
Section 2.2: Figure 2B is confusing; it is not clear which amino acids belong to molecule C and which belong to molecule D. Furthermore, the structure of MAP is not shown, but rather that of MAP-EQRQ.
Section 2.3: Figure 4 is unclear. I assume that the one on the left is Figure 4A and the one on the right is Figure 4B. In Figure 4A, the fragment released by the aniline treatment is barely distinguishable (I assume the arrow indicates the fragment, although this is not indicated anywhere), which is understandable because the electrophoresis was perhaps performed on an agarose gel (the materials and methods do not describe how this was done). It might be easier to distinguish by showing the negative of the picture (dark bands on a white background).
Section 2.4: The role of Y72 and Y118 should be discussed and should also appear in Figure 5B. E168Q is not mentioned in line 166.
Section 2.5: The materials and methods section should explain how the electrophoresis was performed, and the results shown in Figure 6 should be explained in greater detail, indicating the molecular weights of the bands shown and whether these molecular weights correspond to the proteins cited. Indicate what GST is, i.e. glutathione S-transferase.
Lines 195-210: This paragraph does not have sense. The description does not correspond to Figure 7. It needs to be rewritten.
Discussion:
Line 233: The phrase “Up to now, there are no reports of RIPs with RNA.” is unclear.
It would be useful to devote a few lines to the importance or relevance of these experiments (e.g., related to the use of RIPs in biotechnology).
Materials and methods:
Provide a better description of how RNA electrophoresis and SDS-PAGE were performed.
Author Response
Reviewer 2
The authors study the structure of MAP (Mirabilis Antiviral Protein) mutants and how they interact with the sarcin-ricin loop and ribosomal stalk proteins of eukaryotic and prokaryotic ribosomes, comparing it with trichosantin and ricin. MAP, tricosanthin and ricin are ribosome-inactivating proteins (RIPs) that have N-glycosylase activity on the sarcin-ricin loop (SRL) of the major ribosomal RNA. The SRL forms part of the GTPase-associated centre (GAC), which is the landing platform for translational GTPases. RIPs are used in cancer and antiviral therapy, and in agriculture for the construction of transgenic plants resistant to viruses, fungi and insects.
The topic is interesting, and the experiments provide new information that attempts to explain why some RIPs inhibit protein synthesis in bacteria; however, it is not well written, which makes it difficult to understand. Therefore, before being accepted, it needs to be rewritten.
We appreciated the reviewer 3 for all corrections and important suggestions.
Abstract:
Line 8: change “RNA N-glycosidase” by “RNA N-glycosylase”. We revised.
Lines 15 and 73: the expression “active site R171” is confusing.
We changed to “residue R171 at the active site of MAP”.
Introduction:
In order that the reader may understand the purpose of this work, a systematic presentation of the state of the art is required, including (1) a definition of RIPs and types of RIPs, (2) enzymatic activity, (3) the function of the sarcin-ricin loop, (4) the function and composition of the GTPase-associated centre (GAC), and (5) the difference between the GAC of eukaryotes and prokaryotes.
We have arranged the Introduction section clearly to understand our purpose as suggested
L63~L73
During translation on the ribosomes, a large amount of energy is necessary for protein synthesis, which is provided by GTP. The GTP hydrolysis occurs in the GTPase-associated center with the collaboration of the SRL and elongation factors, which are delivered by the eukaryotes ribosome P-stalk composed of a pentameric P-complex, with P0 and two copies of P1/P2 heterodimers[10]. It has been shown that the C-terminal domain (CTD) of ribosomal stalk proteins P1/P2 (P1-CTD, P2-CTD) are responsible for domain-specific binding of elongation factors to deliver them to the GTPase center[11,12].On the other hand, E. coli ribosome stalks are made of L10 and L12 proteins, and the elongation factors bind to the CTD of L12 [21]. The crystal structure of the L12-CTD made of 74 amino acids was reported, but there is no similarity with eukaryotic P1/P2 CTD[22]. No interaction between RIP and the E. coli L10/L12 proteins has been reported.
Lines 21–22, 28, 30, …: change “rRNA N-glycosidase” by “rRNA N-glycosylase”
We corrected all.
Lines 54-55: I am not sure whether it is clear that C11-P2 and C6-P2 are part of the CTD, but it would be useful to indicate this.
We revised as suggestion.
Line 61: Only L10 and L12, or also L11?
Stalk is made of L10 and L12. L11 is part of GTPase-associated center.
Results:
Section 2.1: This section could be improved. There are four invariant amino acids in the active site of ricin: Y80, Y123, E177, and R180. The tyrosines form a sandwich with adenine, and E177 and R180 are the amino acids directly involved in catalysis. They are incorrectly numbered in Figure 1. In addition, the legend for Figure 1 should explain the meaning of the symbols, letters and drawings.
We revised figure 1. We revised the figure legend to explain the meaning of the symbols. We also add the explanation of four invariant amino acids in the active site of ricin in the legend.
Section 2.2: Figure 2B is confusing; it is not clear which amino acids belong to molecule C and which belong to molecule D. Furthermore, the structure of MAP is not shown, but rather that of MAP-EQRQ.
We added “All captions belonged in molecule D. The captions in molecule C are not included to clarify”.
This is the first MAP (MAP-EQRQ) structure, and no wild type MAP structure is not available. Thanks for the reviewer. We revised the summary as follows “Wild type MAP structure has not revealed yet. Here, we expressed, purified, and crystallized the plural recombinant MAPs, including both E168Q and R171Q mutations (MAP-EQRQ) in E. coli, and determined the crystal structure of MAP-EQRQ at 2.1 Å resolution.”
Section 2.3: Figure 4 is unclear. I assume that the one on the left is Figure 4A and the one on the right is Figure 4B. In Figure 4A, the fragment released by the aniline treatment is barely distinguishable (I assume the arrow indicates the fragment, although this is not indicated anywhere), which is understandable because the electrophoresis was perhaps performed on an agarose gel (the materials and methods do not describe how this was done). It might be easier to distinguish by showing the negative of the picture (dark bands on a white background).
A and B are indicated. Thank you for the suggestion. Similar suggestion is obtained from another reviewer, and the picture was replaced better one. However, the aniline treatment seems to be drastic for RNA molecules and often cause their degradation. We have done more than ten times to get clear picture without success. So far, the cleaved short 23S rRNA fragment was visible, then we pursued RT-PCR which compensated the aniline-experiment and showed quantities of depurinated 23S rRNA without aniline treatment.
Section 2.4: The role of Y72 and Y118 should be discussed and should also appear in Figure 5B. E168Q is not mentioned in line 166.
We revised the Figure 5B.
We revised as follows in L293 “The results yielded five predicted structures, which showed that wt-MAP bound uniformly to SRL RNA (Figure 5A). Furthermore, it should be noted that the target adenine flipped out and bound with R171 in all five structures, suggesting that R171 was an essential residue to make a stable complex with SRL (Figure 5B). The adenine position was the same as seen in TRS-adenine complex (1QD2). On the other hand, in MAP E168Q, the adenine shows the same position as wt-MAP in two structures out of five. Interestingly, in MAP R171Q, the different adenine, A2662 were located at the same position in all five structures but the direction of adenine are totally different from one of wt-MAP. Furthermore, in MAP-EQRQ , there were conformational variations among the five predicted conformations of SRL (Figure 5A). “
We also added in figure 5B legend as follows. “Adenine 2660 are stacked using π–π interactions by Y72 and trapped by R171. The distances between NH1(NH2) and adenine N3 are 3.5(3.2) Å.”
Section 2.5: The materials and methods section should explain how the electrophoresis was performed, and the results shown in Figure 6 should be explained in greater detail, indicating the molecular weights of the bands shown and whether these molecular weights correspond to the proteins cited. Indicate what GST is, i.e. glutathione S-transferase.
Thank you for your suggestion, we revised as your suggestion. Regarding to molecular weight, we have not realized that the theoretical molecular wight of MAP/TCS are larger than that of GST (see below). Even though, MAP/TCS run faster than GST in SDS-PAGE. To avoid confusion, detail of the theoretical molecular wight is described in Result section.
i.e. MAP-EQRQ: 257aa MW=28,760, TCS: 253aa MW=27,997, GST: 239aa; MW=27,898, GST-11aa: 237aa; MW=27,511, GST-17aa: 243aa; MW=28,243, GST-74aa: 300aa; MW=33,855.
Lines 195-210: This paragraph does not have sense. The description does not correspond to Figure 7. It needs to be rewritten.
Thanks to reviewer! We corrected the letters on the panels in Figure 7.
Discussion:
Line 233: The phrase “Up to now, there are no reports of RIPs with RNA.” is unclear.
Thank you for the comment. We corrected as suggested by reviewer 2. “Up to now, there are no reports of RIPs in complex with SRL RNA”
It would be useful to devote a few lines to the importance or relevance of these experiments (e.g., related to the use of RIPs in biotechnology).
We added the last sentence as follows in L419.
The antiviral properties of RIPs have been investigated for more than four decades. However, their antiviral mechanisms are still an open question. We expect future studies of RIPs will shed light on the unknown mechanism.
Materials and methods:
Provide a better description of how RNA electrophoresis and SDS-PAGE were performed.
We described details of the RNA electrophoresis and SDS-PAGE in figure legend and Experimental Procedure.

Reviewer 3 Report
Comments and Suggestions for Authors
In this manuscript the authors report the expression and purification of 3 recombinant MAPs in E. coli, including a double mutant (MAP-EQRQ) that lacks inhibitory activity. They also report for the first time the binding of a RIP with the E. coli stalk L12 protein. MAP binds to both the eukaryotic P2-stalk and the E. coli stalk (L12) CTDs. The authors suggest that the inhibition of E. coli ribosome is stalk-dependent. They also suggest that R171 is the essential residue that interacts with the target adenine and forms the MAP-RNA complex before the catalysis.
Therefore, this work can put some light to better understand how RIPs like MAP exhibit potent RNA N-glycosidase activity against E. coli ribosomes. On the other hand, the information presented is new and the conclusions match the data. The images are well presented. However, I found it difficult to read the manuscript in some sections such as 2.4 and 2.6, and some legends of Figures. In addition, there are several errors in the text. English language should be checked. Therefore, there are some concerns which the authors should address before the paper can be accepted.
- Title: change to: Structure of Ribosome-Inactivating Protein from Mirabilis jalapa and Its L12-Stalk-Dependent Inhibition of Escherichia coli ribosome.
- Line 14 - Change to “…using quantitative RT-PCR, we showed that residue R171 at the active site of MAP is a key residue to form the stable complex with the target adenine.
- According to IUBMB Enzyme Nomenclature, RIPs should be "rRNA N-glycosylase" Replace “N-glycosidase” with “N-glycosylase” throughout the text (lines 8, 22, 28, 30, 31, 67, 129, 135, 141, 143,…)
- Line 29, 31. Change “lectin-binding domain” by “lectin domain” or “carbohydrate-binding domain”
- Line 38 - Replace “against an E. coli ribosome” by “against E.coli ribosomes”
- Line 41- It is best to first explain how protein synthesis occurs and then discuss GTP hydrolysis.
- Line 63 - Change to “..amino acids was reported, but there is no similarity with eukaryotic P1/P2 CTD”.
- Line 66 – Change to “….shows inhibitory activity to E. coli ribosomes, it is…”.
- Line 68 -Change to “….,we expressed and purified three recombinant MAPs, …”
- Line 73 – Change to “In addition, we discuss that residue R171 at the active site of MAP is a key residue for forming the stable complex with RNA (SRL).
- Line 76 – Change to “This suggests that MAP inhibition is dependent on the E. coli ribosomal stalk”.
- Figure 1-Correct the numbering in Figure 1 because it does not match the residues E177 and R180. Explain in the legend of the figure what the letters (TT) and all the symbols mean.
- Section 2.2. Explain why there are four molecules in the asymmetric unit, why an asymmetric unit is formed, and what implications this has for ribosome binding.
- Line 112 – Change to “…which are type II RIPs”
- Line 117 – Correct – “Pseudo two fold symmetry was shown as blue”
- Line 121 – Correct- “..shown as bold and other conserved residues”
- It would be interesting to show the structure of the active site of MAP to compare it with MAP-EQRQ.
- Figure 4- Indicate in Figure 4 which is A and which is B. Explain in the legend of the figure that the arrow indicates the RNA fragment released.
- rRNA N-glycosylase assay of MAP and the mutants on E. coli should be included in Experimental Procedure
- Line 133 – Change R171A by R171Q. Correct the sentence- “..whereas those with R171Q, or the EQRQ mutant, did not”.
- Line 139- Correct “.. E168Q at 5 μM (Figure 4B), whereas the double mutant….”
- Line 149- Correct “The amount of intact ribosomal RNA was defined as 1.00 and then standardized using the quantity of 10- and 100-fold diluted samples as 0.1 and 0.01, respectively”.
- Line 168- Explain in text what happens with the E168Q mutant– how is the binding to SRL RNA?
- Line 192- Rewrite the sentence to: “Fifty μg of the MAP-EQRQ or TCS were mixed with 50 μg of each GST-stalk and then the bound proteins were trapped by and eluted from Ni-resin”.
- Line 195- Correct: Peptide binding regions are common in RTA and TCS, however, the direction and mode of binding differs between them.
- Line 196 – Rewrite the sentence to clarify the meaning: “We imposed MAP structure with TCS/RTA with the eukaryotic stalk to see the any differences ….”
- Line 201 – Rewrite the sentence to: “ Both MAP and TCS have hydrophobic properties in the lower half (Figure 7C, F).
- From line 202 to 206: the letters on the panels in Figure 7 do not match those in the text. Panels G and H do not exist in Figure 7.
- Line 233 -Change “Up to now, there are no reports of RIPs with RNA” to “Up to now, there are no reports of RIPs in complex with SRL RNA”
- Line 316 - AlphaFold
There are several errors in the text. English language should be revised.
Author Response
Reviewer 3
In this manuscript the authors report the expression and purification of 3 recombinant MAPs in E. coli, including a double mutant (MAP-EQRQ) that lacks inhibitory activity. They also report for the first time the binding of a RIP with the E. coli stalk L12 protein. MAP binds to both the eukaryotic P2-stalk and the E. coli stalk (L12) CTDs. The authors suggest that the inhibition of E. coli ribosome is stalk-dependent. They also suggest that R171 is the essential residue that interacts with the target adenine and forms the MAP-RNA complex before the catalysis.
Therefore, this work can put some light to better understand how RIPs like MAP exhibit potent RNA N-glycosidase activity against E. coli ribosomes. On the other hand, the information presented is new and the conclusions match the data. The images are well presented. However, I found it difficult to read the manuscript in some sections such as 2.4 and 2.6, and some legends of Figures. In addition, there are several errors in the text. English language should be checked. Therefore, there are some concerns which the authors should address before the paper can be accepted.
We really appreciated the reviewer 2 for corrections of several errors and suggestions.
- Title: change to: Structure of Ribosome-Inactivating Protein from Mirabilis jalapa and Its L12-Stalk-Dependent Inhibition of Escherichia coli ribosome. We revised.
- Line 14 - Change to “…using quantitative RT-PCR, we showed that residue R171 at the active site of MAP is a key residue to form the stable complex with the target adenine. We revised.
- According to IUBMB Enzyme Nomenclature, RIPs should be "rRNA N-glycosylase" Replace “N-glycosidase” with “N-glycosylase” throughout the text (lines 8, 22, 28, 30, 31, 67, 129, 135, 141, 143,…)
We changed all.
- Line 29, 31. Change “lectin-binding domain” by “lectin domain” or “carbohydrate-binding domain”
We changed all.
- Line 38 - Replace “against an E. coli ribosome” by “against E.coli ribosomes”
We revised as your suggestion
- Line 41- It is best to first explain how protein synthesis occurs and then discuss GTP hydrolysis.
The suggested explanation was added in L63 “During translation on the ribosomes, a large amount of energy is necessary for protein synthesis, which is provided by GTP.”.
- Line 63 - Change to “..amino acids was reported, but there is no similarity with eukaryotic P1/P2 CTD”. We revised.
- Line 66 – Change to “….shows inhibitory activity to E. coli ribosomes, it is…”. We revised.
- Line 68 -Change to “….,we expressed and purified three recombinant MAPs, …”We revised.
- Line 73 – Change to “In addition, we discuss that residue R171 at the active site of MAP is a key residue for forming the stable complex with RNA (SRL). We revised.
- Line 76 – Change to “This suggests that MAP inhibition is dependent on the E. coli ribosomal stalk”. We revised.
- Figure 1-Correct the numbering in Figure 1 because it does not match the residues E177 and R180. Explain in the legend of the figure what the letters (TT) and all the symbols mean. We corrected the numbering and added the the legend of the figure what the letters (TT) and all the symbols mean.
- Section 2.2. Explain why there are four molecules in the asymmetric unit, why an asymmetric unit is formed, and what implications this has for ribosome binding.
In the obtained crystal forms, there are four molecules in the asymmetric unit. This is not related to the ribosome binding. To clarify the structure of ribosome-binding, now we are trying to reveal the complex by cryo-EM.
- Line 112 – Change to “…which are type II RIPs” We revised.
- Line 117 – Correct – “Pseudo two fold symmetry was shown as blue” We revised.
- Line 121 – Correct- “..shown as bold and other conserved residues” We revised.
- It would be interesting to show the structure of the active site of MAP to compare it with MAP-EQRQ.
This is the first MAP (MAP-EQRQ) structure, and no wild type MAP structure is not available. Thanks for the reviewer. We revised the summary as follows “Wild type MAP structure has not revealed yet. Here, we expressed, purified, and crystallized the plural recombinant MAPs, including both E168Q and R171Q mutations (MAP-EQRQ) in E. coli, and determined the crystal structure of MAP-EQRQ at 2.1 Å resolution.”
- Figure 4- Indicate in Figure 4 which is A and which is B. Explain in the legend of the figure that the arrow indicates the RNA fragment released.
We revised the figure legend as the suggestion.
- rRNA N-glycosylase assay of MAP and the mutants on E. coli should be included in Experimental Procedure
We described the details in the legend.
- Line 133 – Change R171A by R171Q. Correct the sentence- “..whereas those with R171Q, or the EQRQ mutant, did not”. We revised.
- Line 139- Correct “.. E168Q at 5 μM (Figure 4B), whereas the double mutant….” We revised.
- Line 149- Correct “The amount of intact ribosomal RNA was defined as 1.00 and then standardized using the quantity of 10- and 100-fold diluted samples as 0.1 and 0.01, respectively”. We revised.
- Line 168- Explain in text what happens with the E168Q mutant– how is the binding to SRL RNA?
We revised as follows in L293 “The results yielded five predicted structures, which showed that wt-MAP bound uniformly to SRL RNA (Figure 5A). Furthermore, it should be noted that the target adenine flipped out and bound with R171 in all five structures, suggesting that R171 was an essential residue to make a stable complex with SRL (Figure 5B). The adenine position was the same as seen in TRS-adenine complex (1QD2). On the other hand, in MAP E168Q, the adenine shows the same position as wt-MAP in two structures out of five. Interestingly, in MAP R171Q, the different adenine, A2662 were located at the same position in all five structures but the direction of adenine are totally different from one of wt-MAP. Furthermore, in MAP-EQRQ , there were conformational variations among the five predicted conformations of SRL (Figure 5A). “
- Line 192- Rewrite the sentence to: “Fifty μg of the MAP-EQRQ or TCS were mixed with 50 μg of each GST-stalk and then the bound proteins were trapped by and eluted from Ni-resin”.
We revised as suggested.
Line 195- Correct: Peptide binding regions are common in RTA and TCS, however, the direction and mode of binding differs between them. We revised.
Line 196 – Rewrite the sentence to clarify the meaning: “We imposed MAP structure with TCS/RTA with the eukaryotic stalk to see the any differences ….” We revised.
Line 201 – Rewrite the sentence to: “ Both MAP and TCS have hydrophobic properties in the lower half (Figure 7C, F). We revised.
From line 202 to 206: the letters on the panels in Figure 7 do not match those in the text. Panels G and H do not exist in Figure 7.
Thanks to reviewer! We corrected the letters on the panels in Figure 7.
Line 233 -Change “Up to now, there are no reports of RIPs with RNA” to “Up to now, there are no reports of RIPs in complex with SRL RNA” We revised.
Line 316 – AlphaFold We revised.

Round 2
Reviewer 1 Report
Comments and Suggestions for Authors
The authors have made appropriate revisions.
Author Response
Thank you for your time to review our manuscript.
We believe that the manusciprt was improved significantly.
Reviewer 2 Report
Comments and Suggestions for Authors
The authors have answered the questions I asked them and made corrections that have improved the manuscript. Therefore, the article can be published in its present form.
Author Response

(The authors gave the same response as above.)

Reviewer 3 Report
Comments and Suggestions for Authors
In this revised version of the manuscript, the authors have addressed the comments of the reviewers and have improved the manuscript. The current version has a much more complete content. In my opinion, these changes have significantly improved the clarity of the manuscript. However, there are still several errors in the text. English language should be checked.
Minor comments:
Line 11.- Change by “The structure of the wild type MAP has not been revealed yet”.
Line 79.- Change by “Furthermore, the structure of the wild type MAP is not known yet”.
Line 131.-Change by “The pseudo double symmetry is shown in blue. All captions belong to molecule D. Captions for molecule C are not included for clarity”.
Line 163.- Change “Red allow” by “Red arrow”
Line 185.- Change by “Interestingly, in MAP R171Q, adenine, A2662, was located at the same position in all five structures, but the direction of the adenine was totally different from that of wt-MAP”.
Line 195.- Change by “Adenine 2660 is stacked via π–π interactions by Y72 and trapped by R171”.
Line 210.- Change “molecular wight” by “molecular weight”
Line 223- Change by “We impose the MAP structure with TCS/RTA with the eukaryotic stalk to see if there is any difference in these stalk-binding regions”.
Line 256- Correct the sentence- “In particular, in this case, the search for ……may also make it difficult to obtain the real solution”.
Line 283- Change by “We expect that future studies on RIPs will shed light on this unknown mechanism”.
Line 287- Rewrite the sentence: “The wt-MAP extract was squeezed out from Mirabilis jalapa root, and precipitation obtained by 90% ammonium sulfate saturation was resuspended and dialyzed against …”.
Comments on the Quality of English LanguageThere are several errors in the text. English language should be revised.
Author Response
In this revised version of the manuscript, the authors have addressed the comments of the reviewers and have improved the manuscript. The current version has a much more complete content. In my opinion, these changes have significantly improved the clarity of the manuscript. However, there are still several errors in the text. English language should be checked.
We appreciated for the second comments and corrections of reviewer 3.
We agree that these changes have significantly improved the clarity of the manuscript.
Thank you very much!
Minor comments:
Line 11.- Change by “The structure of the wild type MAP has not been revealed yet”.
We corrected as suggested. L11
Line 79.- Change by “Furthermore, the structure of the wild type MAP is not known yet”.
We corrected as suggested, but the same phrase we used. “The structure of the wild type MAP has not been revealed yet”.L111
Line 131.-Change by “The pseudo double symmetry is shown in blue. All captions belong to molecule D. Captions for molecule C are not included for clarity”.
We believe this phrase is right for crystallography, so it remains as before. “Pseudo two-fold symmetry was shown as blue.” L228
We corrected as suggested. “All captions belong to molecule D. Captions for molecule C are not included for clarity”.”
Line 163.- Change “Red allow” by “Red arrow”
> We corrected. L283
Line 185.- Change by “Interestingly, in MAP R171Q, adenine, A2662, was located at the same position in all five structures, but the direction of the adenine was totally different from that of wt-MAP”.
We corrected as suggested. L378
Line 195.- Change by “Adenine 2660 is stacked via π–π interactions by Y72 and trapped by R171”. >
We corrected as suggested. L394
Line 210.- Change “molecular wight” by “molecular weight” >
We corrected. L409
Line 223- Change by “We impose the MAP structure with TCS/RTA with the eukaryotic stalk to see if there is any difference in these stalk-binding regions”.
> We corrected as suggested. L425
Line 256- Correct the sentence- “In particular, in this case, the search for ……may also make it difficult to obtain the real solution”. > Thanks. We corrected as follows. “Especially, in this case, the search for four molecules in an asymmetric unit may also make it difficult to obtain the real solution.”
Line 283- Change by “We expect that future studies on RIPs will shed light on this unknown mechanism”. We corrected as suggested.L512
Line 287- Rewrite the sentence: “The wt-MAP extract was squeezed out from Mirabilis jalapa root, and precipitation obtained by 90% ammonium sulfate saturation was resuspended and dialyzed against …”. We corrected as follows “ The MAP was extracted from the roots of Mirabilis jalapa. The squeezed solution from the root was precipitated using 90% saturated ammonium sulphate, and then it was dialysed against a 10 mM sodium phosphate buffer (pH 6.0) containing 5 mM 2-mercaptoethanol. Further MAP purification was achieved using cation-exchange chromatography (SP-Sepharose), followed by gel filtration chromatography (Superdex 75).” L515